# Assessing the Intestinal Permeability and Anti-Inflammatory Potential of Sesquiterpene Lactones from Chicory

**DOI:** 10.3390/nu12113547

**Published:** 2020-11-19

**Authors:** Melanie S. Matos, José D. Anastácio, J. William Allwood, Diogo Carregosa, Daniela Marques, Julie Sungurtas, Gordon J. McDougall, Regina Menezes, Ana A. Matias, Derek Stewart, Cláudia Nunes dos Santos

**Affiliations:** 1Instituto de Biologia Experimental e Tecnológica (iBET), Av. República, Qta. Marquês, 2780-157 Oeiras, Portugal; melanie.matos@ibet.pt (M.S.M.); jose.diogo.anastacio@nms.unl.pt (J.D.A.); diogo.carregosa@nms.unl.pt (D.C.); regina.menezes@nms.unl.pt (R.M.); amatias@ibet.pt (A.A.M.); 2CEDOC, Chronic Diseases Research Centre, NOVA Medical School, Faculdade de Ciências Médicas, Universidade NOVA de Lisboa, Campo dos Mártires da Pátria, 130, 1169-056 Lisboa, Portugal; daniela.marques@nms.unl.pt; 3Plant Biochemistry and Food Quality Group, Environmental and Biochemical Sciences, The James Hutton Institute, Dundee DD2 5DA, UK; Will.Allwood@hutton.ac.uk (J.W.A.); Julie.Sungurtas@hutton.ac.uk (J.S.); Gordon.Mcdougall@hutton.ac.uk (G.J.M.); derek.Stewart@hutton.ac.uk (D.S.)

**Keywords:** calcineurin, NFAT, 11β,13-dihydrolactucin, lactucin, lactucopicrin, 11β,13-dihydrolactucopicrin

## Abstract

*Cichorium intybus* L. has recently gained major attention due to large quantities of health-promoting compounds in its roots, such as inulin and sesquiterpene lactones (SLs). Chicory is the main dietary source of SLs, which have underexplored bioactive potential. In this study, we assessed the capacity of SLs to permeate the intestinal barrier to become physiologically available, using in silico predictions and in vitro studies with the well-established cell model of the human intestinal mucosa (differentiated Caco-2 cells). The potential of SLs to modulate inflammatory responses through modulation of the nuclear factor of activated T-cells (NFAT) pathway was also evaluated, using a yeast reporter system. Lactucopicrin was revealed as the most permeable chicory SL in the intestinal barrier model, but it had low anti-inflammatory potential. The SL with the highest anti-inflammatory potential was 11β,13-dihydrolactucin, which inhibited up to 54% of Calcineurin-responsive zinc finger (Crz1) activation, concomitantly with the impairment of the nuclear accumulation of Crz1, the yeast orthologue of human NFAT.

## 1. Introduction

*Cichorium intybus* L. (chicory) has been used for centuries in traditional medicine to treat several ailments, such as liver, kidney and gastrointestinal disorders, diabetes, malaria and other diseases. Indeed, chicory has numerous bioactivities, including antimicrobial, antiparasitic, hepatoprotective and gastroprotective, antidiabetic, antitumoral, analgesic and anti-inflammatory effects [1]. Native to Europe and belonging to the Asteraceae family, it is widely cultivated in countries in Europe, Western Asia, North Africa and the United States. Human consumption of chicory ranges from eating the leaf bulbs to using the roots as a substitute for coffee, which contributes to the intake of several phytochemicals present in this plant. Recently, chicory gained major attention due to the presence of large quantities of health-promoting compounds in its roots, such as inulin, a fructan prebiotic that can be fermented by the colon microbiota promoting health [2,3], and sesquiterpene lactones (SLs), which have substantial bioactive potential [4,5].

SLs are secondary metabolites found in plants. Parthenolide and costunolide are examples of SLs having been described, in previous studies, to possess several medicinal properties. Parthenolide appears to influence tubulin-related functions by binding to tubulin itself while also having anti-proliferative activities [6,7]. Costunolide, besides having anti-proliferative activity, also displays anti-allergenic, anti-microbial, anti-diabetic, as well as neuroprotective bioactivities [8]. Both compounds demonstrated an ability to modulate the inflammatory response by inhibiting the Nuclear factor-κB (NF-κB) pathway [7,8,9]. Some SLs are described to inhibit the phosphorylation of IκB, the inhibitor of NF-κB, thus preventing activation and further translocation of NF-κB to the nucleus [7,8,10]. Their ability to directly interact with the NF-κB subunit p65 (RelA), thereby preventing binding to DNA and consequent transcription of inflammatory cytokines was also demonstrated [9]. Costunolide had the ability to inhibit the production of tumor necrosis factor–alpha (TNF-α) and interleukin-6 (IL-6) in RAW264.7 macrophages [11]. In summary, it is well established that this class of compounds may decrease the inflammatory response, which is a key factor in numerous chronic diseases.

Chicory is rich in several SLs, with the most abundant being lactucin, 11β,13-dihydrolactucin, lactucopicrin and 11β,13-dihydrolactucopicrin. Other SLs are also present in lower quantities, such as 15-oxalyl-lactucin, 15-oxalyl-lactucopicrin, 15-oxalyl-11β,13-dihydrolactucin and 15-oxalyl-11β,13-dihydrolactucopicrin [12]. Unlike costunolide, which is a known intermediate in the terpene biosynthetic pathway in chicory, chicory SLs are still not fully characterized for their potential bioactivities. Lactucin, lactucopicrin and 11β,13-dihydrolactucin have been described to possess analgesic activity [13], and the first two also demonstrated antimalarial activity [14]. Chicory extracts containing these compounds also inhibited the expression and activity of the cyclooxygenase-2 (COX-2) enzyme by inhibiting the NF-κB pathway [15].

Another transcription factor with a fundamental role in inflammatory pathways is the nuclear factor of activated T-cells (NFAT), belonging to the Rel family. NFAT not only regulates immune responses but also other cell functions such as cell cycle progression, migration and angiogenesis [16,17,18]. Once dephosphorylated, NFAT translocates into the nucleus where it requires the presence of different transcription factors to control the expression of several genes. As NFAT can modulate diverse cell functions, these proteins are tightly regulated and maintained inactive in the cytosol [16]. Dysregulation of NFAT activity contributes to the development and/or maintenance of chronic inflammatory and autoimmune diseases. FK506 or tacrolimus is a drug capable of interacting with FK506-binding protein-12 (FKBP12), thus inhibiting the dephosphorylation of NFAT by calcineurin, a serine/threonine phosphatase [19,20]. There are few reports in the literature concerning the anti-inflammatory effect of SLs through interference with the NFAT pathway. However natural extracts from *Arnica montana* containing SLs from the pseudoguaianolide family were reported to inhibit NFAT DNA-binding [21,22].

In the present study we aimed to evaluate the uncharacterized anti-inflammatory potential of chicory SLs. For this purpose, the capacity of the main chicory SLs to permeate the intestinal barrier and become physiologically available was evaluated, and their anti-inflammatory effect was explored in yeast models.

## 2. Materials and Methods

### 2.1. Reagents

All reagents used were of the highest commercially available purity. Parthenolide and costunolide were acquired from Sigma-Aldrich (SML0417, P0667) (Gillingham, UK). Lactucin, lactucopicrin, 11β-13-dihydrolactucin and 11β-13-dihydrolactucopicrin were acquired from Extrasynthese (3809, 3813, 3810, 3811) (Genay Cedex, France).

### 2.2. Permeability of Sesquiterpene Lactones

#### 2.2.1. In Silico Prediction of SLs Permeability

A database containing the 3D structures of ten relevant sesquiterpenes lactones found in chicory (Figure 1) was created using ChembioDraw Software (v.14.0, PerkinElmer, Waltham, MA, USA). Data from these structures were created in mol2 files and imported into Maestro software package (version 2018-4, Schrödinger, New York, NY, USA). The structures were then treated with LigPrep (version 2018-4, Schrödinger) using optimized potentials for liquid simulations (OPLS) forcefield. We defined pH 7.4 ± 2.8 as biologically relevant target pH, using Epik (version 2018-4, Schrödinger). For each molecule, a series of properties and molecular descriptors were calculated using QikProp (version 2018-v4, Schrödinger, New York, USA) [23]. These properties and molecular descriptors are then compared against 95% of known drugs to predict a variety of functions such as human oral absorption and intestinal and brain permeability [23].

#### 2.2.2. Caco-2 Cell Culture

Human colon carcinoma Caco-2 cells (DSMZ, Braunschweig, Germany) were routinely grown in high glucose, high pyruvate, Dulbecco’s modified Eagle medium (DMEM) (Gibco, Life Technologies, Grand Island, NY, USA), supplemented with 10% (*v*/*v*) of heat-inactivated fetal bovine serum (FBS) (Biowest, Nuaillé, France), 100 units/mL penicillin, 100 µg/mL streptomycin (Gibco, Life Technologies, Grand Island, NY, USA) and 10 mM nonessential amino acids (Gibco, Life Technologies, Paisley, UK). Cells were cultured in a humidified atmosphere at 37 °C with 5% CO_2_.

#### 2.2.3. Evaluation of Caco-2 Cell Viability

Cell viability experiments were performed in culture media supplemented with only 0.5% FBS and no antibiotics. Cells were seeded in 96-well TC (tissue culture)-treated microplates at a density of 6.2 × 10^4^ cells/cm^2^ and allowed to reach confluence. Possible cytotoxic effects of the pure SLs on human intestinal epithelial cells (Caco-2) were evaluated using the PrestoBlue^®^ cell viability assay (Invitrogen, Thermo Fisher Scientific, Eugene, OR, USA). Briefly, after reaching confluence, Caco-2 cells were exposed to several concentrations of SLs dissolved in culture medium and incubated at 37 °C and 5% CO_2_ for 4 or 24 h. Cells were then incubated with PrestoBlue^®^ (5% *v*/*v* in culture medium) for 2 h at 37 °C, 5% CO_2_. After this period, fluorescence was measured (Ex./Em. 560/590 nm) in a FLx800 fluorescence microplate reader (BioTek Instruments, Winooski, VT, USA), and cell viability was determined as a percentage of control, after blank subtraction. Fluorescence filters for an excitation wavelength of 560 ± 20 nm and an emission wavelength of 590 ± 20 nm was used. The PrestoBlue^®^ assay is based on the metabolic reduction of resazurin by viable cells into the fluorescent resorufin. The amount of resorufin produced is directly proportional to the number of viable cells.

#### 2.2.4. Permeability across Caco-2 Monolayer

For intestinal permeability assessments of pure SLs, Caco-2 cells were seeded in 12 mm Transwell^®^ inserts (polyester membrane, 0.4 μm pore size, Corning Costar Corp., NY, USA) at a density of 1.0 × 10^5^ cell/cm^2^. Cells were allowed to grow and differentiate to a confluent monolayer for 21 days postseeding with medium change three times per week. This mimics the apical and basolateral sides of the intestinal mucosa. Before the permeability assay, cells were washed with Hank’s balanced salt solution (HBSS) (Gibco, Life Technologies, Paisley, UK), and pure SLs in HBSS were added to the apical side for 4 h at the noncytotoxic concentration of 10 μM, while the basolateral well was replaced by HBSS. The fluorescent marker fluorescein (2.7 µM) was used as control, having been submitted to the same protocol as the SLs. Cells were submitted to an orbital shaking of 125 rpm (Ohaus, Shanghai, China), and samples from the basolateral side were collected after 15 min, 30 min, and then every hour until the end of the assay, and each time the retrieved volume was replaced by fresh HBSS to maintain sink conditions. After the 4 h assay, both apical and basolateral contents were collected, the cells were washed with PBS followed by culture medium, and culture medium was added to both compartments for 24 h to assess monolayer recovery. All samples were stored at −80 °C until LC-MS analysis.

Transepithelial electrical resistance (TEER) of all monolayers was monitored before and during experiments, and 24 h after the assay, to ensure their integrity using an EVOM™ voltmeter (WPI, Berlin, Germany). Before each experiment, TEER was measured and only monolayers with a TEER value higher than 500 Ω cm^2^ were used. Experiments were done in triplicate.

#### 2.2.5. LC-MS Analysis

Ultra High Performance Liquid Chromatography (UHPLC) separations were performed with a Dionex U3000 UHPLC coupled with a U3000 Photo Diode Array (PDA) and an LTQ-Orbitrap XL Mass Spectrometry (MS) system (Thermo-Fisher Ltd., Hemel Hempstead, UK). The MS and PDA systems were tuned and calibrated according to the manufacturers recommended procedures. The UHPLC was operated under Xcalibur and Chromeleon software (Thermo-Fisher Ltd. UK). A flow rate of 300 µL/min was applied, the column and guard (Synergi C18 Hydro-RP 80 Ä, 150 × 2.0 mm, 4 µm particle size; Phenomenex Ltd. U.K.) were maintained at a temperature of 30 °C. The solvent system compositions were, A: HPLC grade water (J.T. Baker, Scientific Chemical Supplies Ltd. UK), and solvent B: HPLC grade acetonitrile (J.T. Baker, Scientific Chemical Supplies Ltd. UK), both acidified with 0.1% MS grade formic acid (Ultima, Fisher Scientific, UK). A sample injection volume of 10 µL was employed in partial-loop mode. The gradient program was as follows: hold 2% B 0–2 min, 2–5% B 2–5 min, 5–45% B 5–25 min, 45–100% B 25–35 min, hold 100% B 35–38 min, 100–2% B 38–39 min, hold 2% B 39–44 min. Autosampler syringe and line washes were performed with 2:8 HPLC grade water: acetonitrile (J.T. Baker, Scientific Chemical Supplies Ltd., UK). The HPLC column eluent was first monitored by the U3000 PDA detector where spectra were collected in wavelength/absorbance mode from 200–600 nm. Additionally, three channel set points were employed, Channel A 210 nm, Channel B 260 nm, Channel C 320 nm. Spectra were collected with a filter bandwidth of 8 nm and wavelength step of 1 nm, the scan rate was 5 Hz.

The UHPLC eluent was next transferred to a Thermo LTQ-Orbitrap XL mass spectrometry system operated under Xcalibur (Thermo-Fisher Ltd. UK). For the first two minutes of analysis the UHPLC eluent was diverted to waste, from two to 39 min the eluent was directed to the MS detector, before being diverted back to waste between 39–44 min. Mass spectra were collected in centroid mode with a primary full scan event (*m*/*z* 100–1000) at a mass resolution of 30,000 (FWHM defined at *m*/*z* 400) within the Fourier Transform (FT) detector. Scan speeds of 0.1 and 0.4 s and automatic gain control of 1 × 10^5^ and 1 × 10^6^ were applied within the Ion Trap (IT) and FT detectors respectively. For LC-MS, the following settings were applied to Electro Spray Ionization (ESI): spray voltage +4.0 kV (ESI+); sheath gas 60; aux gas 30; capillary temperature 280 °C; heated ESI probe temperature 100 °C. Prior to LC-MS analysis, each target sesquiterpene lactone (dissolved in ethanol at 1 µM), was directly infused at 5 µL/min (sheath gas 10; aux gas 5; heated ESI probe temperature 50 °C) and the MS system was tuned to maximize the signal level for each compound. The HPLC chromatogram was next segmented based upon the elution time of each target sesquiterpene lactone, the optimized tune conditions/file for each compound were applied to each segment. With respect to lactucin, 11β,13-dihydrolactucin, 11β,13-dihydrolactucopicrin and lactucopicrin, the optimized tune values were similar and therefore a single optimized tune file was applied from 0–26 min (capillary 44 V, tube lens 110 V, skimmer offset 0 V, multipole RF 400, multipole 00 offset −3.5 V, Lens 0 −4.0 V, multipole 0 offset −5.5 V, lens 1 −11 V, gate lens offset −54.0 V, multipole 1 offset −7.0 V, front lens 6.0 V). With respect to parthenolide (capillary 10 V, tube lens 105 V, skimmer offset 0 V, multipole RF 400, multipole 00 offset −0.75 V, lens 0 −5.0 V, multipole 0 offset −6.0 V, lens 1 −11 V, gate lens offset −50.0 V, multipole 1 offset −9.0 V, front lens 6.25 V) and costunolide (capillary 32 V, tube lens 80 V, skimmer offset 0 V, multipole RF 400, multipole 00 offset −5.5 V, lens 0 −6.0 V, multipole 0 offset −6.25 V, lens 1 −12 V, gate lens offet −80.0 V, multipole 1 offset −19.0 V, front lens 6.75 V), optimised tune files were applied from 26–29 and 29–44 min respectively.

Applying the optimized LC-MS method, the samples were analyzed in a completely randomized order. For the LC-MS analytical block, initially five injections of a quality control sample (QC: equal mix of all experimental samples) were performed for LC-MS system conditioning, the same QC sample was injected three further times, followed by the analysis of nine experimental samples, prior to another QC. Once all experimental samples were analyzed following this pattern, the analytical block was concluded by the analysis of two further QC samples. A control blank sample (HBSS buffer) was analyzed at the start and end of the analytical block. The analytical block was concluded by the collection of a 15-point calibration curve from low-to-high concentration. A 50 µM cocktail of the six sesquiterpene lactones was prepared in HBSS buffer (to match the sample matrix) and serially diluted (50 µM > 25 µM > 12.5 µM > 6.25 µM > 3.125 µM > 1.563 µM > 781.25 nM > 390.625 nM > 195.313 nM > 97.656 nM > 48.828 nM > 24.414 nM > 12.207 nM > 6.104 nM > 3.052 nM). The peaks were integrated and areas obtained within Xcalibur Quan Browser (Genesis algorithm, 11-point Gaussian smoothing) applying the [M+H] ion for each sesquiterpene lactone, other than for parthenolide where the [M+H-H_2_O] ion was applied. The sample and QC peak areas were next quantified against the calibration curves within MS Office Excel.

### 2.3. Anti-Inflammatory Potential of Sesquiterpene Lactones

#### 2.3.1. *Saccharomyces cerevisiae* Strains and Growth Conditions

*S. cerevisiae* strains used in this study are listed in Table 1. The strain YAA5, which encodes the CDRE-*lacZ* report gene, was used for the anti-inflammatory assays. The strain YAA6 was used as negative control.

Strains were maintained in YPD medium [1% (*w/v*) yeast extract (BD Bioscience), 2% (*w/v*) peptone (BD Bioscience), 2% (*w/v*) glucose (Sigma-Aldrich, United States), 2% (*w/v*) agar, pH 6.5] and growth was performed in SC (Synthetic Complete) medium [0.79% (*w/v*) CSM (MP Biomedicals, Inc.—Fisher Scientific, Irvine, CA, USA), 0.67% (*w/v*) YNB (DifcoTM Thermo Scientific Inc., Waltham, MA, USA), and 2% (*w/v*) glucose]. A preinoculum was prepared, and cultures were incubated overnight at 30 °C under orbital shaking. Cultures were diluted in fresh medium, and incubated under the same conditions until the optical density at 600 nm (OD_600_) reached 0.5 ± 0.05 (log growth phase) using the equation: ODi × Vi = (ODf/(2 (t/gt)) × Vf, where ODi = initial optical density of the culture, Vi = initial volume of culture, ODf = final optical density of the culture, *t* = time (usually 16 h), gt = generation time of the strain, and Vf = final volume of culture. Readings were performed in a 96-well microtiter plate using a Biotek Power Wave XS plate spectrophotometer (Biotek^®^ Instruments, Winooski, VT, USA).

#### 2.3.2. Cell Viability

Doses of SLs ranging from 12.5 μM to 100 μM for each compound were used in the yeast viability assays. Cultures obtained were diluted to a final OD_600_ = 0.025 ± 0.0025. The viability assays were performed in 96-wells microplates using 100 µL of diluted culture suspensions and 10 µL of Cell Titer Blue reagent (Promega, WI, USA) according to manufacturer’s instructions for 3 h at 30 °C. Fluorescence was measured in 30 min intervals at emission wavelength 580 nm using the Biotek Power Wave XS Microplate Spectrophotometer (Biotek^®^ Instruments, Winooski, VT, USA).

#### 2.3.3. β-Galactosidase Assays

Cultures at OD_600_= 0.5 ± 0.05 were diluted to OD_600_= 0.1 ± 0.01, transferred to 96-well microplate and challenged with the highest nontoxic concentration previously determined for each compound to investigate their putative inhibitory effect towards the calcineurin-responsive zinc finger (Crz1) activation. Measurements of β-galactosidase activity driven by the calcineurin-dependent response element (CDRE) regulated *lacZ* reporter gene allowed the quantitative assessment of the capacity of each compound to inhibit the activation of Crz1. FK506 (Cayman Chemicals, Ann Arbor, MI, USA) was used as a positive control at a final concentration of 12.5 μM. After 90 min of incubation with the respective compounds at 30 °C under orbital shaking, MnCl_2_ (Merck, Saint Louis, MO, USA) was added at a final concentration of 3 mM. After 90 min of incubation under the same conditions, OD_600_ of cultures were recorded, 10 μL of cell suspensions were transferred to a new 96-well plate followed by addition of 20 μL Y-PER cell lysis reagent (ThermoFisher Scientific), and the plates were incubated for 20 min at 37 °C without agitation. 240 μL of LacZ buffer (8.5 g/L Na_2_HPO_4_ (ROTH), 5.5 g/L NaH_2_PO_4_.H_2_O (Merck), 0.75 g/L KCl (Panreac), 0.246 g/L MgSO4.7H2O (Merck) containing 4 mg/L o-nitrophenyl β-D-galactopyranoside (ONPG) (Sigma–Aldrich^®^—Poole, Dorset, UK) was added to each well, and plates were incubated at 30 °C for 2 h [25]. The OD_420_ and OD_550_ were monitored using a Biotek Power Wave XS Microplate Spectrophotometer (Biotek^®^ Instruments, Winooski, VT, USA). The results were expressed as Miller units (MU) [26], applying the following equation, where V = volume of culture assayed in mL; *t* = reaction time in minutes: Miller unit = 1000 × (OD420 − 1.75 × OD550)/(*t* × V × OD600)(1)

#### 2.3.4. Fluorescence Microscopy 

YAA3 cells were treated with 3.6 µM of 11β,13-dihydrolactucin or 12.5 µM of FK506 for 90 min, followed by incubation with or without 3 mM MnCl2 for further 90 min. On the last 10 min of incubation, 10 µg/mL of Hoechst 33,342 (Sigma) nuclear dye were added. Cells were washed and resuspended in 5 µL of 1,4-Diazabicyclo[2.2.2]octane (DABCO, triethylenediamine) DABCO solution (200 mM DABCO in 75% (*v*/*v*) glycerol, 25% (*v*/*v*) PBS) (Sigma-Aldrich). The preparations were monitored for GFP fluorescence as previously described [27] using a Zeiss Imager Z2 (Zeiss, Germany) fluorescence microscope. Photographs were taken with an Axiocam 506 mono (Zeiss). Three images were taken and analyzed for each sample, each one containing ≈ 600 individual cells. Images were analyzed using Fiji-ImageJ 1.53f (NIH, Bethesda, MD, USA).

#### 2.3.5. Quantitative Real Time PCR

The qRT-PCR analyses were performed as previously described [27]. Briefly, total RNA was extracted using the RNeasy Mini kit (QIAGEN). After cleaning, 4 μg of total RNA was used for reverse-transcription with SuperScript™ II Reverse Transcriptase (Invitrogen). The qRT-PCR was performed in a QuantStudio™ 5 (Applied Biosystems), using SensiFAST™ SYBR Lo-ROX Kit (Bioline) to evaluate expression of the *PMR1* (‘5-CACCTTGGTTCCTGGTGATT-3′; 5′-CCGGTTCATTTTCACCAGTT-3′) (GeneID: 852709), and *GSC2* (5′-CCCGTACTTTGGCACAGATT-3′; 5′-GACCCTTTTGTGCTTTGGAA-3′) (GeneID: 852920) genes. Both *ACT1* (5′-GATCATTGCTCCTCCAGAA-3′; 5′-ACTTGTGGTGAACGATAGAT-3′) and *PDA1* (5′-TGACGAACAAGTTGAATTAGC-3′; 5′-TCTTAGGGTTGGAGTTTCTG-3′) were used as reference genes. The results were expressed as relative mRNA expression levels relative to activation condition (mRNA levels a.u.) of at least three independent biological replicates ± SEM.

#### 2.3.6. Statistical Analysis

Results for the permeability of SLs across Caco-2 monolayers are the averages of three independent experiments and are reported as mean ± SD. Differences amongst the concentrations of pure compounds at the apical side between *t* = 0 and *t* = 4 h were assessed by unpaired *t*-tests (α = 0.05), with Welch’s correction applied whenever variance homogeneity was not confirmed, using the GraphPad Prism 8.4.2 software (GraphPad Software, San Diego, CA, United States).

Results for the yeast cell viability, potential anti-inflammatory effect of all SLs and fluorescence microscopy analyses are the averages of three independent experiments and are reported as mean ± SD. Differences between the controls and the experimental concentrations were assessed by analysis of variance with Dunnett’s multiple comparison tests (α = 0.05) using the GraphPad Prims 8.4.2 software.

The IC_50_ value for 11β,13-dihydrolactucin was also calculated using the GraphPad Prims 8.4.2 software.

## 3. Results

### 3.1. In Silico Prediction of Sls Intestinal Permeability

An in silico approach was conducted using Qikprop to predict the potential intestinal permeability (Table 2) of the eight most abundant sesquiterpene lactones present in chicory, together with their precursor costunolide and the proven anti-inflammatory sesquiterpene lactone parthenolide [28] (Figure 1).

The molecular weights of all sesquiterpenes tested were inside the recommended range [130–725 g/mol] for intestinal permeation (Table 2). Also, within the recommended range were the octanol/water partition coefficients (QPlogPo/w) for all sesquiterpenes between −2 to 6.5. Regarding Polar surface area (PSA), two sesquiterpenes lactones 15-oxalyl-lactucopicrin and 15-oxalyl-11β,13-dihydrolactucopicrin were outside the recommended range. The predicted values for human oral absorption for costunolide and parthenolide were both 100% whereas the remaining sesquiterpene lactones showed an average of 69.83% while the oxalyl conjugates showed an average of 41.19%.

The two models of intestinal permeation (Caco-2 and Madin-Darby canine kidney cells (MDCK) models) showed that costunolide and parthenolide had values above the higher threshold in both models, suggesting substantial predicted permeability. The Caco-2 in silico prediction showed that the oxalyl conjugate sesquiterpene lactones were below the lower threshold suggesting poor predicted permeability, while the remaining sesquiterpene lactones were between both low and high thresholds. However, in the MDCK in silico model, except for costunolide and parthenolide, only lactucin and 11β,13-dihydrolactucin stayed between the thresholds with the remaining sesquiterpene lactones all below the lower threshold. Furthermore, regarding predicted blood–brain barrier permeability, all molecules were within the recommended range, with the exception of 15-oxalyl-11β,13-dihydrolactucopicrin and 15-oxalyl-11β,13-lactucopicrin that were below the recommended range.

### 3.2. In Vitro Permeability of Sls across Caco-2

The six SLs that revealed the most promising results in in silico permeability tests, specifically parthenolide, costunolide, lactucin, lactucopicrin, 11β,13-dihydrolactucin and 11β,13-dihydrolactucopicrin, were selected to proceed to in vitro permeability experiments.

The cytotoxic profile of the compounds was determined in the human intestinal cell line Caco-2 (Figure 2), and noncytotoxic concentrations were selected (i.e., at 10 µM) for permeability testing. Caco-2 cells are a useful model for absorption and transport studies, due to their capacity to spontaneously differentiate into enterocytelike cells with expression of tight junctions and microvilli [29].

Transepithelial electrical resistance (TEER) of the cell monolayers was measured before (*t* = 0 h) and after the experiment (*t* = 4 h), as well as 24 h after the end of the assay (Figure 3). Compared to what was seen for fluorescein, which is commonly used as a standard for transport in transepithelial permeability in Caco-2 [30,31], there was a decrease in TEER during the first 4 h of incubation with all SLs, reflecting a reduction in cohesion of tight junctions. When the cells are further maintained in culture medium for 24 h after the permeability assay, there was a recovery of TEER values to similar levels as those obtained before the experiment.

Samples were collected from the apical side at the beginning of the experiment and after 4 h, and then analyzed by LC-MS. The LC-MS method was optimized for SL detection and quantification and based upon the calibration curves obtained in a sample-matched matrix (HBSS buffer), we defined the range of linearity, limit of detection (LOD) and limit of quantification (LOQ) for costunolide, parthenolide, lactucin, lactucopicrin, 11β,13-dihydrolactucin and 11β,13-dihydrolactucopicrin (Appendix A). A statistically significant decrease (*p* < 0.05) of the concentration of SLs in the apical side of the Caco-2 monolayer was observed for all compounds except 11β,13-dihydrolactucin, which suggests removal of SLs possibly across the cell monolayer or into the cells (Table 3). Parthenolide, costunolide and lactucopicrin were reduced to concentrations lower than the LOQ in each case. On the other hand, both lactucin and 11β,13-dihydrolactucopicrin were reduced by ≈30%. In contrast, 11β,13-dihydrolactucin only displayed a concentration decrease of 8.7% after 4 h.

Components with *m*/*z* values and exact mass derived formulae consistent with the presence of SL conjugates with cysteine were detected in the apical side of the cell monolayer after the 4 h experiments (i.e., for parthenolide, costunolide, lactucin and lactucopicrin; electrospray ionization (ESI) positive mode mass spectra are presented in Appendix A). These putative cysteine conjugates were considerably more polar, eluting before their parent SLs. Since standards for the SL-cysteine conjugates are not available, these species were estimated against the calibration curves of their parent SLs (Table 4).

The cysteine-SL conjugates detected by LC-MS for parthenolide, costunolide, lactucin and lactucopicrin displayed apparent higher ionization efficiencies than the parent SLs which led to much higher MS ion intensities and overestimation of the conjugate concentration (Table 4). This may occur because protonation can also occur on the cysteine residue. In the case of lactucin, the estimated concentration for the cysteine conjugate was lower than the concentration of the parent compound, probably because of the low uptake rate of this SL. Therefore, the absolute quantification of the total amount of each SL, both parent and cysteine-conjugates, was not possible.

### 3.3. Anti-Inflammatory Potential of SLs

The cytotoxic profile of SLs in yeast was defined and the highest nontoxic concentration of SLs identified (Figure 4). For every SL tested, concentrations higher than 12.5 µM showed significant cytotoxicity. Costunolide was proven to be even more cytotoxic towards yeast cells, with the highest nontoxic concentration being 6.25 µM, half of the value recorded for other compounds.

Chicory SLs were evaluated regarding their anti-inflammatory potential towards NFAT using the anti-inflammatory drug FK506 as a positive control due to its known inhibitory actions towards calcineurin. As depicted in Table 5, the SL with the highest anti-inflammatory potential was 11β,13-dihydrolactucin, displaying 43% reduction of Crz1 activity. Other SLs failed to significantly decrease Crz1 activity.

#### 3.3.1. Anti-Inflammatory Potential of 11β,13-Dihydrolactucin

A dose response assay, using the same yeast reporter system, was performed to assess the minimal concentration of 11β,13-dihydrolactucin necessary to significantly reduce Crz1 activation. 11β,13-dihydrolactucin inhibited Crz1 activation by ~ 54% at 3.6 µM (Figure 5), and the calculated IC_50_ was 2.35 ± 1.45 μM. To the best of our knowledge, this is the first report of potential inhibitory effects of pure SLs towards NFAT.

#### 3.3.2. 11β,13-Dihydrolactucin Modulates Crz1 Nuclear Accumulation

To characterize further the effect of 11β,13-dihydrolactucin on Crz1 activity, fluorescence microscopy was used as a means to assess the nuclear translocation of the transcription factor. As depicted in Figure 6, Crz1 was dispersed throughout the cells in the control condition. Challenging cells with MnCl_2_ triggered Crz1 nuclear accumulation (55.6%), which was inhibited by the treatment of cells with 11β,13-dihydrolactucin (21%) in a similar manner to that of the positive control FK506, known to inhibit calcineurin activity and thereby preventing the release of Crz1 to the nucleus. These data strongly suggest 11β,13-dihydrolactucin as a novel anti-inflammatory lead for attenuation of the human Crz1 orthologue, NFAT.

#### 3.3.3. 11β,13-Dihydrolactucin Inhibits the Expression of Crz1 Target Genes

We monitored the mRNA levels of Crz1-regulated genes by qPCR, specifically *PMR1* and *GSC2*, encoding a high affinity Ca^2+/^Mn^2+^ P-type ATPase and a cell wall biosynthetic enzyme, respectively. In agreement with the inhibitory effect of 11β,13-dihydrolactucin on Crz1 activity, treatment of cells with this compound downregulated the expression of *PMR1* and *GSC2* to levels comparable to that FK506 (Figure 7). These data further support the role of 11β,13-dihydrolactucin in the regulation of Crz1 activity.

## 4. Discussion

Chicory is a major dietary source of SLs but, unlike other SLs from other sources [6,7,8], they have not been assessed for anti-inflammatory potential. However before considering any bioactivity, the physiological availability of the compounds must be evaluated. In this study, we assessed the capacity of SLs to permeate the intestinal barrier based on in silico predictions and in vitro studies using differentiated Caco-2 cells, the well-established cell model of the human intestinal mucosa.

From the in silico approach conducted using Qikprop, the potential intestinal permeability of the SLs present in chicory could be predicted. Whether sesquiterpenes can reach the entero-hepatic circulation and eventually the systemic circulation may depend to some extent on the physicochemical structural properties of each molecule. The molecular weights of all SLs were similar, with only lactucopicrin, 11β,13-dihydrolactucopicrin, and the oxalyl conjugates having higher molecular masses. Indeed, all compounds were within the recommended molecular weight range (130–725 g/mol) for intestinal permeation. Also, all SLs had octanol/water partition coefficients and solubility figures in the middle of the range for predicted permeability. On the other hand, the polar surface area values of 15-oxalyl-11β,13-dihydrolactucin and 15-oxalyl-lactucin were in the higher half of the recommended range, while 15-oxalyl-lactucopicrin and 15-oxalyl-11β,13-dihydrolactucopicrin were outside the recommended range, suggesting that oxalate conjugation would decrease passive permeability. The values for percentage of human oral absorption revealed that the oxalyl conjugates had predicted values of absorption around 40%, about 30% less than their parent SL. This compares with 100% predicted absorption for costunolide and parthenolide.

The intestinal permeation in Caco-2 and MDCK models was also predicted using Qikprop. The oxalyl conjugates were below the threshold of 25, predicting poor transport across both Caco-2 and MDCK models. The parent SLs gave values between the thresholds of 25 (poor) and 500 (high) in both models. By contrast, costunolide and parthenolide displayed values above 500, suggesting high permeability in both models. Furthermore the predicted blood–brain barrier permeability, one of the most tightly controlled and highly impermeable barriers in human physiology, showed all molecules were inside the range of recommended values except for compounds 15-oxalyl-11β,13-dihydrolactucopicrin and 15-oxalyl-11β,13-lactucopicrin.

Overall, many chicory SLs were predicted to be capable of crossing membranes and potentially reaching the blood stream. Nevertheless, oxalyl conjugates revealed poor predicted permeability due to their increased polar surface in comparison with their respective non-oxalylated structures.

The permeability of the SLs was also evaluated in an in vitro model of the human intestinal mucosa. The four different paths for intestinal absorption are paracellular diffusion, transcellular diffusion, transcytosis and carrier-mediated transport [29]. The decrease in transepithelial electrical resistance (TEER) of the cell monolayers observed during the first 4 h for all SLs reflects a reduction in cohesion of tight junctions, which is consistent with paracellular transport through the intercellular space [29]. This result is comparable to that of fluorescein, which is standardly used as a control for transepithelial permeability in Caco-2 cells [30,31]. Nonetheless, the observed decrease in TEER does not indicate toxicity, since these values returned to pre-experiment levels when the cells were maintained for 24 h after the permeability assay, implying the recovery of tight junction functionality.

The in silico predictions for costunolide and parthenolide for the percentage of human oral absorption and the QPPCaco-2 results agreed with the observed in vitro permeability data, but other molecules showed different patterns. Indeed, Qikprop predicted that lactucin and 11β,13-dihydrolactucin should have higher permeability than lactucopicrin and 11β,13-dihydrolactucopicrin, probably due to their lower molecular weights and PSA, both of which are important parameters for membrane permeability and bioavailability [32]. The differences between the in silico and in vitro determinations of SL permeability could be due to the presence of additional biological mechanisms in the cells that were not considered in silico, such as cellular metabolism or membrane transporters. In fact, several influx and efflux transporters present in Caco-2 membranes may affect the transport of compounds belonging to the SL class [33]. Moreover, slight structural differences among SLs are known to completely alter their permeability across Caco-2 monolayers [34], which could influence the very different in vitro permeability of the SLs (i.e., from 8% to 100% decrease in apical levels). In particular, lactucopicrin has a pronounced tridimensional shape whereas the other chicory SLs are more planar [35]. Three-dimensionality of chemical structures is an important feature for drug-likeness, by contributing to increased solubility and permeability of molecules, as well as molecular recognition by biological targets including membrane transporters [36,37].

Nevertheless, since the in vitro method only evaluated the decrease of the SLs on the apical site, the metabolism of compounds cannot be discarded. In fact, the identification of cysteine conjugates of some SLs on the apical side of Caco-2 suggests that these SLs were either formed by enzymatic action within the cell, followed by efflux through membrane transporters back to the apical side, or underwent a nonenzymatic conjugation with cysteine. Both mechanisms have already been described in Caco-2 cells for nobilin, a sesquiterpene lactone of the germacranolide type [38]. In fact, Caco-2 cells express the transporters P-gp, BCRP, MRP1, MRP2, MRP3, and OATP as well as phase II conjugating enzymes including glutathione S-transferase and UDP-glucuronosyltransferase [39]. The extensive bioconversion of nobilin by Caco-2 cells results in the formation of three conjugation products, i.e., with glucuronic acid, cysteine, and glutathione [38]. Based on these reports, the SL-cysteine species detected may result from a glutathione conjugate formed intracellularly then hydrolyzed by the brush border membrane enzymes, γ-glutamyltranspeptidase and/or dipeptidases, yielding the cysteine conjugate, a process already described for other glutathione conjugates and also observed for nobilin [38,40]. Alternatively, a direct conjugation with cysteine by a nonenzymatic reaction with the SLs might have taken place. The α-methylene-γ-lactone moiety and other α,β-unsaturated carbonyl structures of sesquiterpene lactones are known to react by Michael-type addition with cysteine and cysteine-containing molecules, including glutathione [41]. Indeed, costunolide can spontaneously form cysteine and GSH conjugates even in the absence of glutathione-S-transferase (GST) [42], which opens up the possibility of other SLs displaying similar behavior. GSH has a role in the intracellular detoxification and elimination of xenobiotics by forming conjugates that can then be exported from the cell by multidrug resistance proteins (MRPs) known to be active in both small intestine and colon tissues [43,44]. Although intracellular concentrations of cysteine are low, there is evidence that exogenous compounds can also directly form conjugates with free cysteine intracellularly [45] and these conjugates may also be transported to the extracellular medium through MRPs [44]. Extracellular cysteine conjugation is highly unlikely for several reasons; cysteine is rapidly oxidized to cystine in the extracellular environment [46], thus becoming unavailable to react with SLs due to the loss of the free thiol group; transport of free cysteine to the apical extracellular medium is not likely to occur as cysteine limits the rate of GSH synthesis due to its low intracellular concentration [46]. Overall, the SL-cysteine conjugates are probably the result of SL uptake, followed by its conjugation with GSH or free cysteine, and efflux of the conjugate to apical media.

Notably, cysteine adducts were not detected for 11β,13-dihydrolactucin or 11β,13-dihydrolactucopicrin. A possible explanation is the lack of the α-methylene-γ-lactone moiety in their structures since this group is largely responsible for the reactivity of SLs with sulfhydryl groups [41]. Notably, as no SL-cysteine adducts were formed for these two compounds, their respective concentration decline in the apical side after the 4 h experiment, although not pronounced, probably results from actual uptake or transport of the compounds.

No cysteine conjugates for parthenolide or other SLs were reported in previous absorption studies in Caco-2 cells using HPLC [47,48]. However, this may reflect the targeted nature of the previous HPLC-based studies and highlights the advantages of the LC-MS method developed here. Concerning lactucin, the low amounts of the cysteine-conjugate compared to the other SLs could reflect that lactucin was not as well taken up by cells, and therefore less conjugate was produced.

Overall, we observed differential in vitro absorption and evidence of bioconversion of the main chicory SLs and to the best of our knowledge, there are still no reports about their in vivo bioavailability. Indeed, little is known about the ADME of chicory SLs and SLs in general, with the few published studies with Caco-2 cells suggesting that sesquiterpene lactones are well absorbed by diffusion [47,48,49] and undergo carrier mediated efflux and influx [38].

The potential of SLs to modulate inflammatory responses through modulation of nuclear factor of activated T-cells (NFAT) pathway was also evaluated, using a yeast reporter system, based on the activity of Crz1. Both in human and yeast, NFAT and Crz1, respectively, are under the control of calcineurin. Human calcineurin and yeast calcineurin share a high evolutionary similarity making this model ideal for the screening of compounds with anti-inflammatory potential [50]. Like in mammalian cells, in the presence of a stimulus the yeast calcineurin dephosphorylates the transcription factor Crz1 enabling its translocation into the nucleus, thus promoting the expression of CDRE-regulated genes [25].

11β,13-dihydrolactucin significantly decreased the activity of Crz1, displaying the strongest anti-inflammatory potential of the chicory SLs. This specific activity of 11β,13-dihydrolactucin may arise as, unlike other SLs, it does not effectively form cysteine conjugates and is available to interact with target proteins. Indeed, the capacity of SLs to react with molecules containing cysteine is reported to be mainly due to the presence of α,β-unsaturated carbonyl groups with the presence of an α-methylene-γ-lactone ring enhancing the reaction rate [51]. 11β,13-dihydrolactucin lacks the α-methylene-γ-lactone ring and therefore may not readily form these conjugates.

Since 11β,13-dihydrolactucin yielded the most promising anti-inflammatory potential in the yeast reporter assay, we explored further the potential of this SL. A dose-response assay was performed to assess the concentration providing protective activity in the presence of a stimulus. The low IC_50_ value for Crz1 inhibition by 11β,13-dihydrolactucin suggests that the low absorption rate observed may still provide the potential for modulation of Crz1 activity. The ability of 11β,13-dihydrolactucin to modulate Crz1 activity is very significant and comparable to the well described immunosuppressant drug FK506, known for inhibition of calcineurin and also already described to be active in the yeast system [27].

The Crz1-GFP fusion protein was explored to investigate the capacity of 11β,13-dihydrolactucin to inhibit Crz1 activation by preventing its nuclear accumulation. It was noteworthy that treatment of cells with 11β,13-dihydrolactucin prior to induction with MnCl_2_ strongly reduced the percentage of cells displaying nuclear-located Crz1-GFP (21%) as compared to cells treated only with MnCl_2_ (55.6%). Altogether, the data support the hypothesis that 11β,13-dihydrolactucin prevents the nuclear accumulation of Crz1, thereby modulating reporter gene expression through the CDRE-response element. This modulation is what is determining the reduced β-galactosidase activity observed in 11β,13-dihydrolactucin-treated cells before activation. In addition, 11β,13-dihydrolactucin downregulated endogenous Crz1 targets, further supporting its role in the modulation of Crz1, and potentially NFAT, activity.

Similar results were obtained in cells treated with the immunosuppressant drug FK506, also known as tacrolimus, which blocks calcineurin function. Tacrolimus binds to FKBP12 protein, blocking the catalytic site of calcineurin, thus inhibiting the activation of the NFAT and Crz1 transcription factors. However, the use of FK506 blocks calcineurin activity, which is proved to play crucial roles especially in neurological and nephrotic tissues [52,53]. Thus, it is essential to elucidate the mechanisms by which 11β,13-dihydrolactucin modulates Crz1/NFAT activity. Finding a drug that selectively targeted NFAT could avoid the side effects of blocking calcineurin activity. Further studies should focus on the molecular targets of 11β,13-dihydrolactucin, both in yeast and mammalian cell models, to clarify if it could avoid the secondary effects of direct blocking calcineurin activity.

## 5. Conclusions

Chicory *(Cichorium intybus*) is one of the major sources of dietary SLs, the activity of which is largely understudied. In silico predictions suggested that some SLs may be able to reach blood stream. Using the Caco-2 cell model, we determined the in vitro permeability of four abundant chicory SLs: lactucin, lactucopicrin, 11β,13-dihydrolactucin and 11β,13-dihydrolactucopicrin, as well as their precursor costunolide and the well-known SL, parthenolide. Lactucopicrin was the most permeable chicory SL in this model system, however, cysteine adducts of lactucopicrin and lactucin were detected by LC-MS, suggesting bioconversion of SLs by the cells. Further analysis revealed that the SL with the highest potential to inhibit Crz1 activation was 11β,13-dihydrolactucin, suggesting that its anti-inflammatory potential was achieved through the inhibition of NFAT. Calcineurin is highly conserved among eukaryotes and the docking of NFAT sequence recognized by calcineurin is highly similar to Crz1. This makes the yeast model suitable for studying the interaction of compounds with the mammalian equivalent calcineurin-NFAT pathway. Therefore, this study is the first of its kind in assessing the ability of SLs to modulate NFAT-associated inflammatory responses.

Altogether, our data revealed that 11β,13-dihydrolactucin could be an alternative anti-inflammatory compound since its effect was similar to that of the drug FK506, preventing the activation of the Crz1 transcription factor through retention in the cytosol. However, further studies are crucial to validate the findings in other advanced cellular models and to address the molecular targets of 11β,13-dihydrolactucin, particularly to clarify if this specific SL exerts a direct effect on Crz1/NFAT without affecting calcineurin activity, opposed to what is known for FK506.

## Figures and Tables

**Figure 1 nutrients-12-03547-f001:**
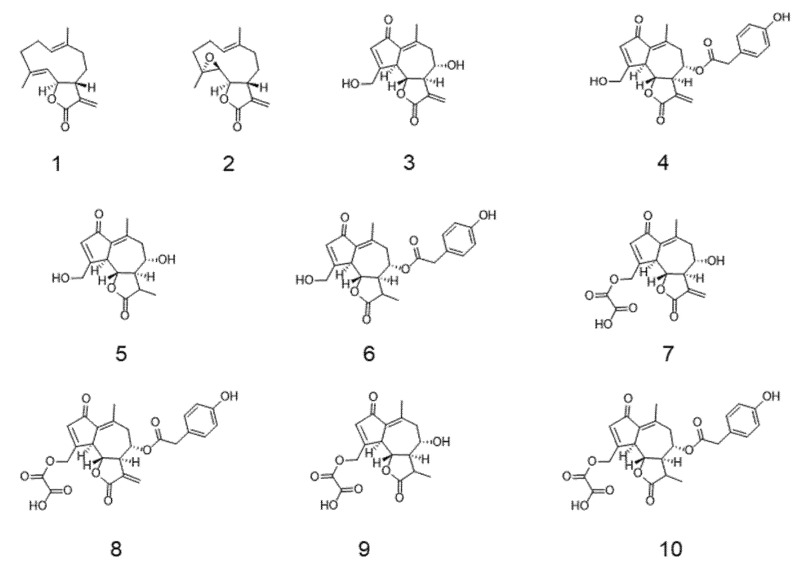
Chemical structure of tested sesquiterpene lactones (SLs). (**1**) Costunolide; (**2**) parthenolide; (**3**) lactucin; (**4**) lactucopicrin; (**5**) 11β,13-dihydrolactucin; (**6**) 11β,13-dihydrolactucopicrin; (**7**) 15-oxalyl-lactucin; (**8**) 15-oxalyl-lactucopicrin; (**9**) 15-oxalyl-11β,13-dihydrolactucin; (**10**) 15-oxalyl-11β,13-dihydrolactucopicrin.

**Figure 2 nutrients-12-03547-f002:**
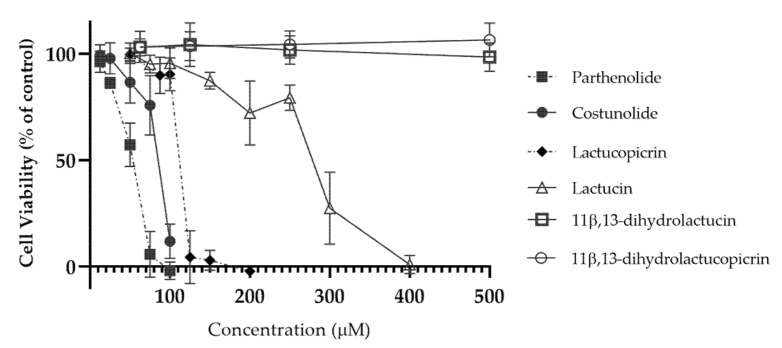
Cytotoxic profile of parthenolide, costunolide, lactucopicrin, lactucin, 11β,13-dihydrolactucin and 11β,13-dihydrolactucopicrin in Caco-2 cells. Cytotoxicity was evaluated using the PrestoBlue^®^ cell viability assay, by testing the compounds between 12.5 and 500 μM. Data are presented as means ± SD, *n* = 3.

**Figure 3 nutrients-12-03547-f003:**
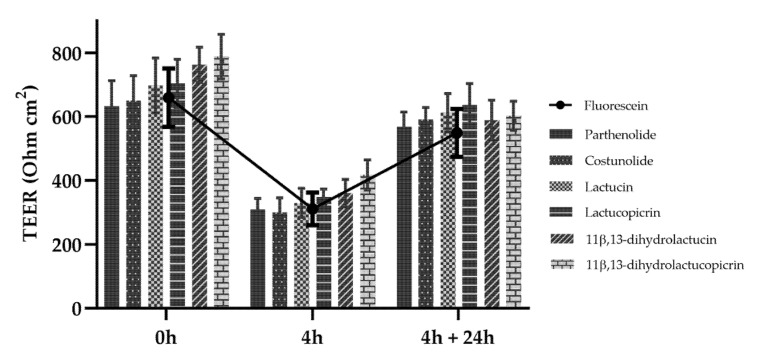
Transepithelial electrical resistance (TEER) of Caco-2 cells. TEER measurements were made before and after the 4 h permeability experiment and at the 24 h recovery period after the end of the assay. SLs were tested at a concentration of 10 μM and fluorescein at 2.7 μM. Data are presented as means ± SD, *n* = 3.

**Figure 4 nutrients-12-03547-f004:**
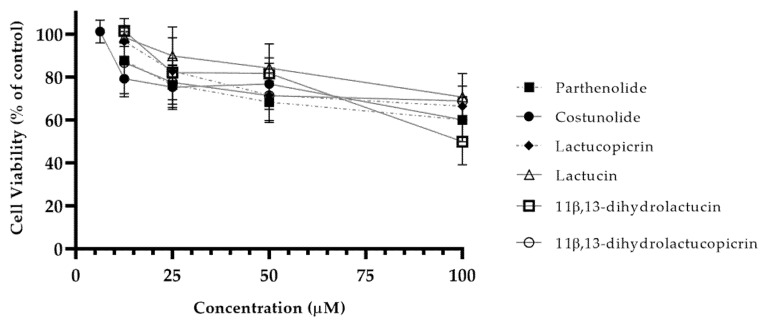
Cytotoxicity of chicory SLs in yeasts. Yeast cells were incubated with different concentrations (6.25–100 µM) of SLs to determine the highest nontoxic concentration. Cell viability for each compound was determined using the Cell Titer Blue assay. Data are presented as means ± SD, *n* = 3.

**Figure 5 nutrients-12-03547-f005:**
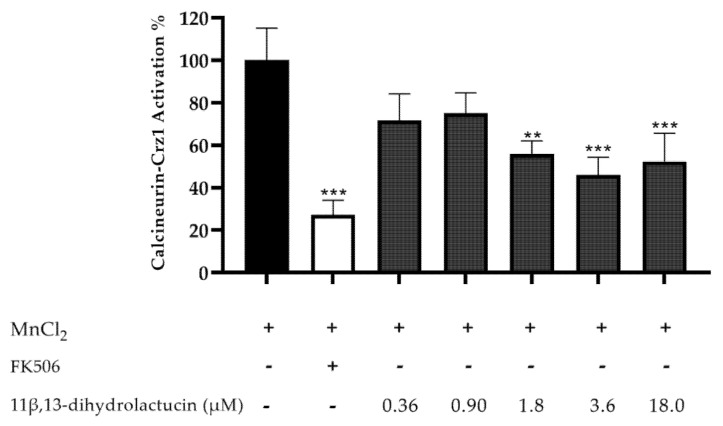
Anti-inflammatory potential of 11β,13-dihydrolactucin as demonstrated by the inhibition of the calcineurin-Crz1 pathway in *S. cerevisiae*. Statistical differences are noted as ** *p* < 0.01, *** *p* < 0.001 relative to the activated control (cells treated with MnCl_2_).

**Figure 6 nutrients-12-03547-f006:**
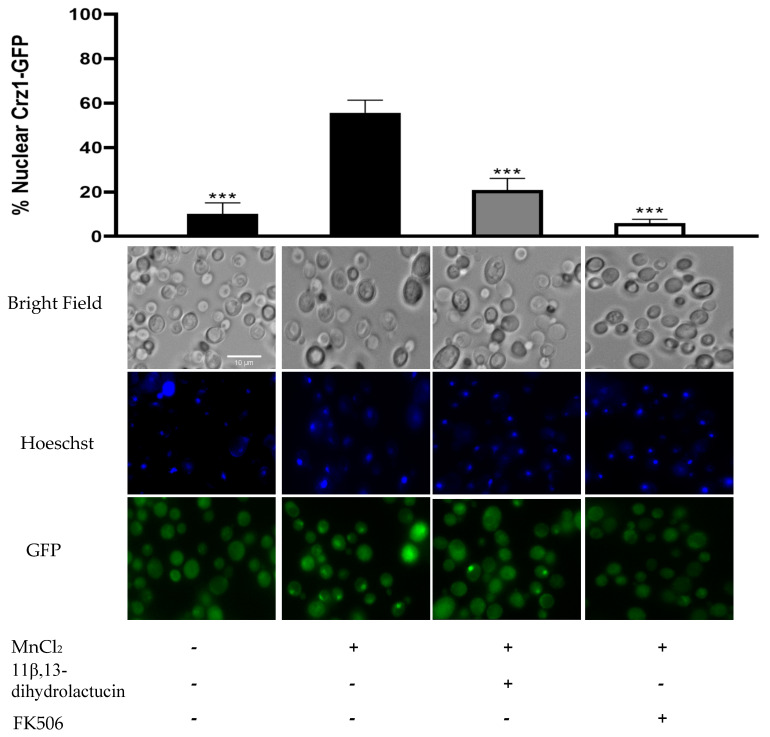
11β,13-dihydrolactucin inhibits Crz1-GFP nuclear accumulation. YAA3 cells were first treated or not with 3.6 µM of 11β,13-dihydrolactucin, challenged with 3 mM MnCl_2_ and Crz1-GFP subcellular distribution was monitored using fluorescence microscopy. The immunosuppressant FK506 was used as a positive control. Approximately 1800 cells for each condition were counted. Representative imagens are shown, and the values represent the mean of percentage of nuclear Crz1-GFP ± SD of three biological replicates. Statistical differences are denoted as *** *p* < 0.001 relative to activated cells.

**Figure 7 nutrients-12-03547-f007:**
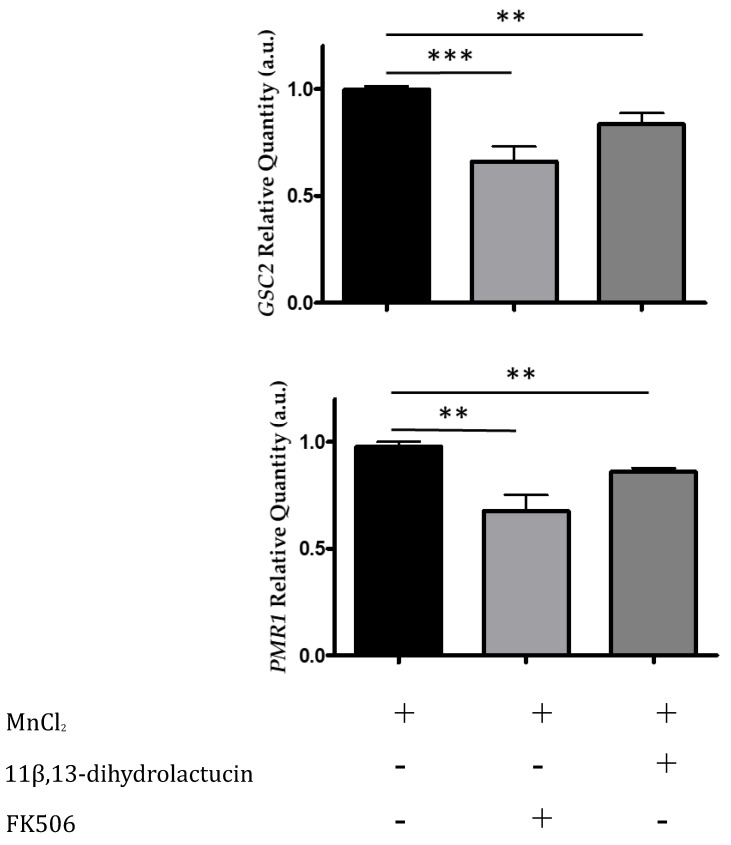
11β,13-dihydrolactucin modulates the downregulation of endogenous Crz1 target genes. BY4741 cells were subjected to 3.6 µM of 11β,13-dihydrolactucin, induced with 3 mM MnCl_2_ and the mRNA levels of Crz1 target genes *PMR1* and *GSC2* were assessed by qPCR. The immunosuppressant FK506, which inhibits calcineurin and prevents Crz1 activation, was used as a positive control. The values represent the mean ± SEM of at least three biological replicates, ** *p* < 0.01, *** *p* < 0.001.

**Table 1 nutrients-12-03547-t001:** List of *S. cerevisiae* strains used in this study.

Strain	Genotyping Information	Reference
YAA3 (BY4742-Crz1-GFP)	*his3::CRZ1-GFP-HIS3*	[24]
YAA5 (BY4742-CDRE-lacZ)	*aur1::AUR1-C-4xCDRE-lacZ*	[24]
YAA6 (BY4742-crz1_CDRE-lacZ)	*aur1::AUR1-C-CDRE-lacZ Δcrz1::KanMX4*	[24]

**Table 2 nutrients-12-03547-t002:** In silico QikProp predicted parameters of passive membrane permeation for different SLs.

Compound	MW ^1^	QP logPo/w ^2^	QP logS ^3^	PSA ^4^	% Human Oral Absorption ^5^	QP PCaco-2 ^6^	QP PMDCK ^7^	QP logBB ^8^
Costunolide	232.32	2.67	−2.97	40.49	100.00	2402.79	1276.02	0.01
Parthenolide	248.32	1.82	−1.94	52.84	100.00	2712.92	1454.94	0.06
Lactucin	276.29	0.10	−2.08	106.60	68.61	198.54	86.21	−1.19
Lactucopicrin	410.42	1.56	−3.76	141.91	68.04	61.11	24.12	−2.04
11β,13-dihydrolactucin	278.30	0.18	−2.30	105.89	70.09	224.82	98.60	−1.07
11β,13-dihydrolactucopicrin	412.44	1.79	−3.70	140.56	72.56	95.54	39.21	−1.68
15-oxalyl-lactucin	348.31	−0.03	−2.45	170.30	40.99	6.42	2.69	−2.23
15-oxalyl-lactucopicrin	482.44	1.48	−4.47	204.80	39.36	1.96	0.76	−3.42
15-oxalyl-11β,13-dihydrolactucin	350.32	0.08	−2.65	169.25	43.07	7.61	3.23	−2.07
15-oxalyl-11β,13-dihydrolactucopicrin	484.46	1.50	−4.38	203.57	41.32	2.06	0.79	−3.11
Acceptable ranges	130–725	−2–6.5	−6.5–0.5	7–200	−	−	−	−3–1.2

^1^ MW: Molecular weight (g/mol) (130 ≤ MW ≤ 725). ^2^ QPlogPo/w: Predicted octanol/water partition coefficient (−2 QPlogPo/w 6.5). ^3^ QPlogS: Predicted aqueous solubility (−6.5 < QPlogS < 0.5). ^4^ PSA: Van der Waals surface area of polar atoms (7 < PSA < 200). ^5^ % Human Oral Absorption. ^6^ QPPCaco: Predicted apparent Caco-2 cell permeability in nm/sec (QPPCaco < 25—poor absorption; QPPCaco > 500—great absorption). ^7^ QPPMDCK: Predicted apparent Madin-Darby canine kidney cells (MDCK) cell permeability in nm/sec (QPPMDCK < 25—poor absorption; QPPMDCK > 500—great absorption). ^8^ QPlogBB: Predicted brain/blood partition coefficient (−3 < QPlogBB < 1.2).

**Table 3 nutrients-12-03547-t003:** Concentration of SLs in the apical compartment before and after the permeability experiment.

Compound	Concentration of SL (Apical Side) (µM)		% of Decrease in SL Concentration in the Apical Side (*t* = 4h)
	*t* = 0 h	*t* = 4 h	
Costunolide	13.1 ± 1.8	<LOD	100 **
Parthenolide	11.9 ± 2.0	<LOQ	100 **
Lactucin	8.3 ± 0.7	5.9 ± 0.5	28.8 ***
Lactucopicrin	12.4 ± 1.3	<LOQ	100 **
11β,13-dihydrolactucin	9.1 ± 1.3	8.3 ± 1.4	8.7
11β,13-dihydrolactucopicrin	7.0 ± 1.2	4.9 ± 0.9	29.9 *

Statistical differences are noted as * *p* < 0.05, ** *p* < 0.01 and *** *p* < 0.001. Statistical significance refers to % SL decrease after the 4h experiment compared to their concentration at *t* = 0 h.

**Table 4 nutrients-12-03547-t004:** Comparison between SLs and their respective cysteine conjugates, in terms of retention time, *m*/*z* and peak areas. Parent SLs were added at 10μM in the apical side at *t* = 0h.

Compound	RT (min) ^a^	*m*/*z*^b^	Peak Area (Average a.u.)	Apical Concentration *t* = 4 h (µM Equivalents to Parent SL)
Costunolide	31.30	233.153	-	<LOD
Costunolide-Cys	20.70	354.173	3.1 × 10^8^	95 ± 8.4
Parthenolide	27.44	231.137	9.2 × 10^5^	<LOQ
Parthenolide-Cys	17.14	370.168	4.6 × 10^8^	27 ± 5.9
Lactucin	13.82	277.106	6.8 × 10^7^	5.9 ± 0.5
Lactucin-Cys	9.83	398.126	3.5 × 10^7^	2.9 ± 3.6
Lactucopicrin	22.20	411.142	1.2 × 10^6^	<LOQ
Lactucopicrin-Cys	16.10	532.162	2.4 × 10^8^	17.8 ± 2.9

^a^ RT: Retention time in minutes. ^b^
*m*/*z*: Mass:charge ratio.

**Table 5 nutrients-12-03547-t005:** Inhibition of calcineurin-Crz1 pathway by different SLs.

Compound	Concentration (µM)	Calcineurin-Crz1 Inhibition (%) ± SD
FK506	12.5	67 ± 13 ***
Parthenolide	12.5	20 ± 12
Costunolide	6.25	18 ± 9
Lactucopicrin	12.5	18 ± 9
Lactucin	12.5	20 ± 14
11β,13-dihydrolactucin	12.5	43 ± 8 ***
11β,13-dihydrolactucopicrin	12.5	26 ± 11

Statistical differences are noted as *** *p* < 0.001 relative to control. FK506 = positive control.

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
