# Peer review of "Assessing the Intestinal Permeability and Anti-Inflammatory Potential of Sesquiterpene Lactones from Chicory"

_nutrients, 2020, doi:10.3390/nu12113547_

Round 1
Reviewer 1 Report
Authors used sesquiterpene lactones from chicory to assess their anti-inflammatory potential. In silico prediction and in vitro analyses were performed. The subject is very interesting and well explained in the introduction.
However there are many points to improve:
Lines 83-89: There are conclusions of the article. Authors must reformulate here their objectives only.
Line 93: Please replace “Costunolide” by “costunolide”
Line 259: I suggest to separate Results from Discussion
Lines 305-311: There is only a setup of LC-MS parameters. This part is not a result. I suggest to remove from the article and eventually to include in a supplementary file.
Line 313: Please replace “namely” by “respectively” or other reformulation.
Line 316: The cytotoxic profile is announced but there are not results presented. Authors must include a table with results.
Lines 317-325: These explanations must be removed and presented in Material and methods.
Concerning results for in vitro permeability, authors must present: cytotoxicity results (table to include), TEER results (table to include) and LC-MS results (only tables 4 and 5). Please remove Figs 2-5.
Lines 327-333, 346-362 and 447-487 must be added to Discussion section.
Line 488: This section must be reorganized. Lines 489-500 must be includes in Material and methods.
Line 496: “their highest non-toxic concentrations” are mentioned but these results are not presented. Authors must be careful to include this in results (as mentioned before). Similarly, in the table 6 there are announced SLs concentrations but no analyze indicates that the announced amounts are toxic or not and why partenolide is at the half of others.
Lines 515-519: This discussion is too speculative. Why obtained results are comparable to FK drugs (no comparison and no citation are provided) ? This is not an argument to claim that NFAT and Crz1 transcription factors are repressed. No qRT-PCR data are presented to sustain this affirmation.
Author Response
Rebuttal letter
Please find enclosed the revised manuscript entitled ”Assessing the intestinal permeability and anti-inflammatory potential of sesquiterpene lactones from chicory” by Melanie S. Matos, José D. Anastácio, William J. Allwood, Diogo Carregosa, Daniela Marques, J. Sungurtas, G. J. McDougall , Regina Menezes, Ana A. Matias, Derek Stewart and myself, Cláudia Nunes dos Santos, to be considered for publication in Nutrients. First, the authors would like to acknowledge the editor and the reviewers for their useful contributions to improve the quality of this manuscript. In this revised manuscript, we considered the suggestions made by the reviewers. Thus, the goal of this letter is to describe the changes made in the manuscript and discuss some of the issues addressed by the reviewers.
Reviewer 1
Authors used sesquiterpene lactones from chicory to assess their anti-inflammatory potential. In silico prediction and in vitro analyses were performed. The subject is very interesting and well explained in the introduction.
However there are many points to improve:
Lines 83-89: There are conclusions of the article. Authors must reformulate here their objectives only.
These sentences have been removed from the introduction as requested and placed in the Conclusion section.
Line 93: Please replace “Costunolide” by “costunolide”
Replaced as requested.
Line 259: I suggest to separate Results from Discussion
Results and Discussion have been separated into two different sections respectively.
Lines 305-311: There is only a setup of LC-MS parameters. This part is not a result. I suggest to remove from the article and eventually to include in a supplementary file.
This part is now in the supplementary files as suggested.
Line 313: Please replace “namely” by “respectively” or other reformulation.
Replaced as requested.
Line 316: The cytotoxic profile is announced but there are not results presented. Authors must include a table with results.
The graphs regarding the cytotoxic profile of compounds were added for both Caco-2 cells (figure 2) and yeast cell model (figure 4).
Lines 317-325: These explanations must be removed and presented in Material and methods.
These explanations were removed and added to the Materials and methods sections as requested.
Concerning results for in vitro permeability, authors must present: cytotoxicity results (table to include), TEER results (table to include) and LC-MS results (only tables 4 and 5). Please remove Figs 2-5.
Cytotoxicity results (figure 2), as well as TEER results (figure 3) were added to the main text. Figures 2-5 (chromatograms) were removed from the main text and added to the supplementary material file.
Lines 327-333, 346-362 and 447-487 must be added to Discussion section.
These paragraphs are now under the Discussion Section as requested.
Line 488: This section must be reorganized. Lines 489-500 must be included in Material and methods.
Both suggestions were done: section reorganized and suggested lines were integrated in Materials and Methods section.
Line 496: “their highest non-toxic concentrations” are mentioned but these results are not presented. Authors must be careful to include this in results (as mentioned before). Similarly, in the table 6 there are announced SLs concentrations but no analyze indicates that the announced amounts are toxic or not and why parthenolide is at the half of others.
Cytotoxicity profile was added (Figure 4). We had an error regarding the names of costunolide and parthenolide in the wrong lines, which has been corrected. As you can appreciate from the cytotoxicity profile, costunolide demonstrates cytotoxicity at 12.5 µM, which is why we did the bioactivity assay using 6.25 µM.
Lines 515-519: This discussion is too speculative. Why obtained results are comparable to FK drugs (no comparison and no citation are provided) ? This is not an argument to claim that NFAT and Crz1 transcription factors are repressed. No qRT-PCR data are presented to sustain this affirmation.
FK506 was used as a positive control, the results obtained for the compound 11β,13-dihydrolactucin were directly compared in the assay. Text was rewritten to clarify the comparison and a reference added as requested.
Moreover to reinforce the role of Crz1 and its repression by 11β,13-dihydrolactucin in comparison to FK506 ewe have evaluated the Crz1 nuclear translocation by fluorescence microscopy using a fusion of Crz1-GFP, that clearly reveal the impairment of Crz1 nuclear translocation. Moreover we also perform as requested qRT-PCR for target genes described to be under the control of Crz1, to demonstrate a reduction of gene expression promoted by 11β,13-dihydrolactucinBoth sets of data strengthen the comparison between the results obtained using the compound 11β,13-dihydrolactucin and the drug FK506.
Reviewer 2 Report
This is a good quality and original scientific article. The abstract and the purpose of the study are clearly summarized. The statistical methods are valid and correctly applied. The reference list covers the relevant literature adequately.
Below some remarks and corrections :
-The biactive properties of sesquiterpene lactons are evaluated excluding those which have oxalate conjugates. The authors justify this choice because of increased polar surface of oxalate conjugates in comparison with their respective non-oxalate structures. In the same way, when no SL-cysteine adducts were formed for 11β,13-dihydrolactucin and 11β,13-dihydrolactucopicrin, some structural arguments were also developed.
But, the authors don’t try to explain the differences observed between lactucopicrin and 11β,13-dihydrolactucin concerning the permeability and the antiinflammatory activities by their structures. It can be done.
-The authors should briefly argue about why the sesquiterpene lactone parthenolide was used in the assays for bioactivities in addition to costunolide. They have already mentioned that costunolide is the precursor of SLs, but it is also useful to explain the relevance of using parthenolide in this study.
-A reference is needed at the end of the conclusions when authors compared the anti-inflammatory effect of 11β,13-dihydrolactucin with the known drug FK506 (lines 542-546). Th other names of FK506 should be mentioned in brackets: (tacrolimus, fujimycin)
-Some corrections are needed in the text editing :
60, 61: write "oxalyle" instead of "oxallyl"
18, 33, 488, 528: When the sentences or titles are written in italic type, the latin names of chicory must be in Roman type and the "L." after the name is in italic type like the rest of the sentences or titles. Vice versa, when the italic names are used in Roman type written sentences the latin name of the plant must be in italic type.
83-89: There is no need to announce the results in the introduction. The authors only need to clearly explain the objective. I think these sentences can be deleted.
94: remove "-" after "11" and before "β" (2 times)
192: replace "was" by "were" in "...initially five injections of....was performed"
223: be careful about "3.2. Cell viability" because there is no "3.1." before. It should be "2.3.2" and not in italic type
231: write "β-galactosidase assays" instead of "β-. galactosidase assays"
488: write "..intybus L." instead of "...intybus.L"
507: remove "." after "-" in "11béta,13-. Dihydrolactucin"
Author Response
Rebuttal letter
Please find enclosed the revised manuscript entitled ”Assessing the intestinal permeability and anti-inflammatory potential of sesquiterpene lactones from chicory” by Melanie S. Matos, José D. Anastácio, William J. Allwood, Diogo Carregosa, Daniela Marques, J. Sungurtas, G. J. McDougall , Regina Menezes, Ana A. Matias, Derek Stewart and myself, Cláudia Nunes dos Santos, to be considered for publication in Nutrients. First, the authors would like to acknowledge the editor and the reviewers for their useful contributions to improve the quality of this manuscript. In this revised manuscript, we considered the suggestions made by the reviewers. Thus, the goal of this letter is to describe the changes made in the manuscript and discuss some of the issues addressed by the reviewers.
Reviewer 2
This is a good quality and original scientific article. The abstract and the purpose of the study are clearly summarized. The statistical methods are valid and correctly applied. The reference list covers the relevant literature adequately.
Below some remarks and corrections:
-The biactive properties of sesquiterpene lactones are evaluated excluding those which have oxalate conjugates. The authors justify this choice because of increased polar surface of oxalate conjugates in comparison with their respective non-oxalate structures. In the same way, when no SL-cysteine adducts were formed for 11β,13-dihydrolactucin and 11β,13-dihydrolactucopicrin, some structural arguments were also developed.
But, the authors don’t try to explain the differences observed between lactucopicrin and 11β,13-dihydrolactucin concerning the permeability and the antiinflammatory activities by their structures. It can be done.
Some structural arguments were presented in discussion concerning the differential permeability and anti-inflammatory potential of SLs.
-The authors should briefly argue about why the sesquiterpene lactone parthenolide was used in the assays for bioactivities in addition to costunolide. They have already mentioned that costunolide is the precursor of SLs, but it is also useful to explain the relevance of using parthenolide in this study.
Parthenolide is a model SL with proven anti-inflammatory potential. This was added to section 3.1 of the results to clarify the criteria of its use.
A reference is needed at the end of the conclusions when authors compared the anti-inflammatory effect of 11β,13-dihydrolactucin with the known drug FK506 (lines 542-546). Th other names of FK506 should be mentioned in brackets: (tacrolimus, fujimycin)
FK506 was used as a positive control, the results obtained for the compound 11β,13-dihydrolactucin were directly compared in the assay. Text was rewritten to clarify the comparison and a reference added as requested.
Moreover to reinforce the role of Crz1 and its repression by 11β,13-dihydrolactucin in comparison to FK506 we have evaluated the Crz1 nuclear translocation by fluorescence microscopy using a fusion of Crz1-GFP, that clearly reveal the impairment of Crz1 nuclear translocation. Moreover we also perform as requested qRT-PCR for target genes described to be under the control of Crz1, to demonstrate a reduction of gene expression promoted by 11β,13-dihydrolactucin. Both sets of data strengthen the comparison between the results obtained using the compound 11β,13-dihydrolactucin and the drug FK506.
Other names for FK506 such as tacrolimus have been added to the text in the discussion section.
-Some corrections are needed in the text editing :
60, 61: write "oxalyle" instead of "oxallyl"
18, 33, 488, 528: When the sentences or titles are written in italic type, the latin names of chicory must be in Roman type and the "L." after the name is in italic type like the rest of the sentences or titles. Vice versa, when the italic names are used in Roman type written sentences the latin name of the plant must be in italic type.
Corrected as requested.
83-89: There is no need to announce the results in the introduction. The authors only need to clearly explain the objective. I think these sentences can be deleted.
These sentences have been removed from the introduction as requested.
94: remove "-" after "11" and before "β" (2 times)
Corrected as requested.
192: replace "was" by "were" in "...initially five injections of....was performed"
Corrected as requested.
223: be careful about "3.2. Cell viability" because there is no "3.1." before. It should be "2.3.2" and not in italic type
All section titles were revised and corrected.
231: write "β-galactosidase assays" instead of "β-. galactosidase assays"
Replaced as requested.
488: write "..intybus L." instead of "...intybus.L"
Corrected as requested
507: remove "." after "-" in "11béta,13-. Dihydrolactucin"
Corrected as requested.
Round 2
Reviewer 1 Report
The article is much improved. I thank the authors for considering my advices.
I have still identified typing errors (e.g. line 533 capital letter) but overall the structure and writing are very well done.